# Predicting asthma attacks in primary care: protocol for developing a machine learning-based prediction model

Holly Tibble,[1,2] Athanasios Tsanas,[1,2] Elsie Horne,[1,2] Robert Horne,[2,3] Mehrdad Mizani,[1,2] Colin R Simpson,[2,4] Aziz Sheikh[1,2]

[1]Usher Institute of Population Health Sciences and Informatics, Edinburgh Medical School, College of Medicine and Veterinary Medicine, University of Edinburgh, Edinburgh, UK
[2]Asthma UK Centre for Applied Research, Edinburgh, UK
[3]University College London, London, UK
[4]School of Health, Victoria University of Wellington, Wellington, UK

**Correspondence to**
Holly Tibble;
holly.tibble@ed.ac.uk

## ABSTRACT

**Introduction** Asthma is a long-term condition with rapid onset worsening of symptoms ('attacks') which can be unpredictable and may prove fatal. Models predicting asthma attacks require high sensitivity to minimise mortality risk, and high specificity to avoid unnecessary prescribing of preventative medications that carry an associated risk of adverse events. We aim to create a risk score to predict asthma attacks in primary care using a statistical learning approach trained on routinely collected electronic health record data.

**Methods and analysis** We will employ machine-learning classifiers (naïve Bayes, support vector machines, and random forests) to create an asthma attack risk prediction model, using the Asthma Learning Health System (ALHS) study patient registry comprising 500 000 individuals across 75 Scottish general practices, with linked longitudinal primary care prescribing records, primary care Read codes, accident and emergency records, hospital admissions and deaths. Models will be compared on a partition of the dataset reserved for validation, and the final model will be tested in both an unseen partition of the derivation dataset and an external dataset from the Seasonal Influenza Vaccination Effectiveness II (SIVE II) study.

**Ethics and dissemination** Permissions for the ALHS project were obtained from the South East Scotland Research Ethics Committee 02 [16/SS/0130] and the Public Benefit and Privacy Panel for Health and Social Care (1516–0489). Permissions for the SIVE II project were obtained from the Privacy Advisory Committee (National Services NHS Scotland) [68/14] and the National Research Ethics Committee West Midlands–Edgbaston [15/WM/0035]. The subsequent research paper will be submitted for publication to a peer-reviewed journal and code scripts used for all components of the data cleaning, compiling, and analysis will be made available in the open source GitHub website (https://github.com/hollytibble).

## INTRODUCTION

Asthma is a long-term lung disease characterised by inflammation of the airways, which may manifest as episodic wheezing, chest tightness, coughing, and shortness of breath. An asthma attack is the sudden worsening of symptoms, which may prove fatal.[1] In 2017, asthma was estimated to affect 235 million

### Strengths and limitations of this study

► This analysis is based on a large, representative dataset comprising over 500 000 individuals recruited from 75 general practices across Scotland.
► We will employ novel applications of established machine learning and training data enrichment methodologies.
► The prediction model we develop will be tested in unseen large external dataset, namely the SIVE II dataset.
► This derivation and validation work will be undertaken in NHS Scotland; there will therefore be a need for further validation work in other UK nations and international contexts.

people worldwide.[2] In 2015 alone, 1434 people died from asthma attacks in the UK—a rate of 2.21 deaths per 100 000 person-years.[3] Asthma attack incidence is reported to be between 0.01 and 0.78 events per person-year, depending on the definition of attacks and the population (eg, primary care, secondary care).[4–6]

Asthma therapy typically follows a fairly linear path—beginning with a short-acting bronchodilator in the individuals without persistent asthma symptoms and adding preventative treatments and long-acting bronchodilators in the individuals with more persistent asthma symptoms.[7 8] The individuals with persistent troublesome symptoms and/or considered to be at very high risk may be prescribed biologicals and/or oral steroids.[9] Oral steroids are often considered a last resort due to their undesirable safety profile including increased risk of diabetes,[10–12] osteoporosis,[13–15] and affective and psychotic disorders.[15–18]

It follows that the determination of those at high risk for asthma attacks is crucial in order to prevent attacks and minimise the risk of unnecessary side effects. Furthermore, the 2014 National Review of Asthma

Deaths found that 45% of asthma deaths in the study year occurred without the patient requesting medical help or before help could be provided.[5] Increased awareness of the risk could prevent the patients with asthma from delay in seeking medical care and preventing fatality.

While it might seem intuitive that the patients with most severe daily symptoms exhibit greater risk of severe morbidity and mortality, research suggests that these symptoms may be a suboptimal clinical marker of asthma attack risk.[19] Indeed, some people with asthma are more prone to asthma attacks than others, with asthma attack history being the strongest risk factor for future asthma attacks.[20–23] Other commonly identified risk factors for asthma attacks include poor asthma control[24–27] (often a result of poor adherence to preventative therapy[28–31]), smoking,[24 27 32–34] history of hospital admission,[21 24] history of oral steroid use,[24] obesity,[27 34–38] access to medicines,[39 40] socioeconomic status,[41 42] and viral respiratory infections.[43–45]

Despite the identification of many risk factors, identifying high-risk individuals has proven a challenging task. Logistic regression, the most commonly used statistical method for event prediction, is known to predict outcomes poorly when there is *class imbalance* (event and no event),[46] and we expect the problem investigated in this study assessing asthma attacks will be highly imbalanced. For example, a model could predict that a very rare event would never occur, and it would be correct in the vast majority of cases. As such, most prediction models report high *specificity* (correctly predicting low attack risk to those who did not have attacks), but low *sensitivity* (correctly predicting high risk in those who did go on to have attacks),[4 24 41 47–51] which results in less reliable risk prediction for patients at high risk.

In a recent study by Finkelstein and Jeong,[52] sensitivity (and specificity) in excess of 75% was achieved for all classifiers (adaptive Bayesian network, naïve Bayes classifier, and support vector machine) predicting asthma attacks a week in advance using a sample of just over 7000 records of home tele-monitoring data. They found substantial improvements in model sensitivity using *training enrichment* methods, pre-processing the training data to improve the performance in the testing data—for example, by increasing the prevalence of the rare outcome in the training data to balance the classes.

## RESEARCH AIM

We aim to create a personalised risk assessment tool to assist primary care clinicians in predicting asthma attacks over a period of 1, 4, 12, 26, and 52 weeks, employing machine-learning methodologies such as naïve Bayes classifiers, random forests, and support vector machines, as well as ensemble algorithms. The model will build on previous research[4 24 41 47–52] to improve the sensitivity of our event prediction, without unduly compromising the specificity. This is crucial in order to reduce prescribing steroid, diminish the long-term effects of high steroid use over a lifetime, which have adverse effects,[10–18] and reduce patient anxiety when risk of an asthma attack is low.

Primary care consultations provide the opportunity for patients and clinicians to assess changes to asthma attack risk, which can be used to promote patients to seek emergency care if there is a significant deterioration in their symptoms and to promote risk-reducing lifestyle choices.

## METHODS

### Data sources and permissions

The derivation dataset used for training, validating, and testing the model will be the Asthma Learning Healthcare System (ALHS) dataset, created in order to develop and validate a prototype learning health system for asthma patients in Scotland.[53] The ALHS study aims to increase understanding of variation in asthma outcomes and create benchmarks for clinical practice in order to reduce suboptimal care by repurposing patient data to create a continuous loop of knowledge generation, evidence-based clinical practice change, and change assessment. The study dataset contains patient demographics from the patient registry, primary care prescribing records, primary care encounters, Accident and Emergency (A&E) records, hospital inpatient admissions and deaths, linkable by an anonymised unique identifier. Datasets were extracted between November 2017 and August 2018 for the period January 2000 to December 2017, as shown

**Table 1** Metadata for clinical data sources in derivation dataset (ALHS)

| Data Source | Number of Records | Number of Individuals | Extraction Date | Data Specification Date Range |
|---|---|---|---|---|
| Primary Care Prescribing* | 4 709 231 | 47 095 | October 2018 | January 2009–April 2017 |
| Primary Care Encounters* | 11 766 100 | 49 307 | March 2018 | January 2000–November 2017 |
| Accident & Emergency | 1 831 789 | 500 321 | November 2017 | June 2007–September 2017 |
| Hospital Inpatient Admissions | 1 668 957 | 342 838 | August 2018 | January 2000–March 2017 |
| Mortality | *NA* | 91 758 | May 2018 | January 2000–March 2017 |

*Records available for subset of study population with asthma diagnosis only.

**Table 2** Metadata for clinical data sources in external dataset (SIVE II)

| Data Source | Number of Records | Number of Individuals | Extraction Date | Data Specification Date Range |
|---|---|---|---|---|
| Primary Care Prescribing | 29 360 448 | 1 073 377 | May 2017 | January 2003–March 2017 |
| Primary Care Encounters | 31 878 423 | 1 887 957 | May 2017 | January 2000*–March 2017 |
| Accident & Emergency | 4 116 561 | 1 247 314 | April 2017 | June 2007– August 2016 |
| Hospital Inpatient Admissions | 3 549 174 | 794 937 | April 2017 | January 2000–March 2017 |
| Mortality | NA | 215 466 | April 2017 | January 2000– March 2017 |

*Diagnosis codes entered in this period, but post-dated from 1940 onwards retained.

in table 1, along with the number of records and unique individuals before data cleaning.

In order to verify that the prediction model performance is not limited to the development dataset and that it generalises well in new, unseen data presented to the classifier in the training process, we will evaluate its performance using an external cohort study dataset, the second Seasonal Influenza Vaccination Effectiveness (SIVE II) cohort study,[54 55] which used a large national primary care (1.25 million individuals from 230 Scottish general practices) and laboratory-linked dataset to evaluate live attenuated and trivalent inactivated influenza vaccination effectiveness. The SIVE II dataset contains records from the same sources (primary and secondary care) and modalities (diagnosis and date) as the ALHS dataset (extraction and specification dates are shown in table 2), and can be harmonised such that variables and value sets are aligned. In Appendix A, we detail the data harmonisation plan, that is, we list the key variables to be used in the following analyses, their format in each dataset (for example, whether age is pre-coded into 5-year bands), and the common denominator format that will be used in the analyses to ensure the highest degree of concordance during the validation stage.

### Patient and public involvement

This analysis plan was constructed with the assistance of the Asthma UK Centre for Applied Research (AUKCAR) Patient and Public Involvement (PPI) group. The particular importance of avoiding a substantial decrease in specificity in order to gain higher sensitivity was a result of discussions within this group about the burden of side effects from preventative treatment.

### Inclusion criteria

We will identify our study population as all adults (aged 18 and over) with asthma being identified by clinical diagnoses (Read codes), without a chronic obstructive pulmonary disease (COPD) diagnosis, and with relevant prescribing records in primary care. Patients with missing sex or age information will be removed; this and any other patient exclusions from further analysis will be explicitly detailed.

All records from the derivation dataset (ALHS) will be left-censored on January 2009 in order to align with

the primary care prescribing data and right-censored on March 2017 in order to align with the mortality, primary care, and inpatient hospital admission records, as presented in table 1. Similarly, records from the external dataset (SIVE II) will be left-censored on January 2003 in order to align with the primary care prescribing data and right-censored on August 2016 to align with the A&E records, as shown in table 2. There is a high probability that some individuals will have been recruited into both studies, and therefore those individuals will be flagged in the external testing dataset and removed from the study pool.

### Outcome ascertainment

We will identify asthma attacks, defined by the American Thoracic Society/European Respiratory Society,[56] as a prescription of oral corticosteroids, an asthma-related A&E visit, or an asthma-related hospital admission. Additionally, deaths occurring with asthma as the primary cause will be labelled as asthma attacks. Instances of multiple attack indicators occurring within a 14-day period were coded as a single attack.

### Patient characteristics, confounders, and missing data

Patient characteristics at baseline will be reported and included as time-varying confounders in analyses. For all characteristics derived from Read codes, full code lists will be provided as online supplementary materials.

*Demographics*: Age, sex, rurality, and social deprivation will be extracted from the primary care registry. Social deprivation is measured using quintiles of the Scottish Index of Multiple Deprivation (SIMD), a geographic measure derived using data on income, employment, education, health, access to services, crime, and housing.[57] Rurality is defined using the Scottish Government Urban Rural Classification Scale (6-fold scale).[58] While missing age and/or sex are exclusion criteria for the study sample, absence for rurality and social deprivation will be coded as 'missing.'

*Practice Location*: Practice location will be included in order to account for clustering of patients by region. Location will be coded using the Nomenclature of Territorial Units for Statistics[59] (NUTS 3) codes, linked from the registered practice data zone (2001).

*Asthma Severity*: Asthma severity will be categorised using the British Thoracic Society's 2016 5-step treatment classification.[60] Severity will be considered time-dependent and will be determined using prescribing records at any change in regimen.

*Smoking Status*: Smoking status will be derived from primary care data and presented as a 3-level variable, namely current, former, and non-smoker, using the most recent smoking Read code at any day. Smoking status will be considered time-dependent and determined using the most recent Read code records, and the individuals with unknown smoking status will be coded as non-smokers.[61 62]

*Blood Eosinophil Count:* Blood eosinophil count will be derived from primary care Read codes and will be dichotomised at ≥400 cells/µL. The individuals with non-recorded blood eosinophil count will be coded as missing. Blood eosinophil count will be considered time-dependent and determined using the most recent Read code record.

*Obesity*: Obesity will be derived from body mass index (BMI) or height and weight records in primary care data and will be presented as a binary variable (BMI≥30). The individuals with unknown BMI will be coded as non-obese. Obesity will be considered time-dependent and determined using the most recent Read code record.

*Comorbidity*: Comorbidity will be defined by 17 dichotomous (unweighted) variables representing the diagnostic categories of the adapted Charlson Comorbidity Index.[63 64] Additionally, active diagnoses of rhinitis, eczema, gastro-oesophageal reflux disease, nasal polyps, and anaphylaxis will be recorded; all identified by Blakey *et al* as contributing characteristics to increased asthma attack risk.[65] Comorbidities will be considered time-dependent and determined using all prior Read code records.

*Previous Healthcare Usage*: The number of repeat prescriptions of preventer medication and the number of primary care asthma encounters (days on which at least one asthma related code was recorded) in the previous year will be derived from primary care prescribing and Read code records, respectively. Both will be considered time-dependent and determined using records from the previous calendar year.

*Asthma Control:* The mean short-acting beta-2 agonist dose per day will be estimated retroactively by examining the dates between prescriptions. The most recent peak expiratory flow measurement at any time will be recorded (categorical, based on percentage of previous maximum) or coded as missing if that measurement was more than 7 days ago. Adherence to preventer therapy will be approximated using the medication possession ratio,[66] calculated from primary care prescribing records.

*History of Asthma Attacks*: Prior asthma attacks will be identified solely using primary care prescribing records and Read codes. This is because primary care practitioners will not be able to make use of secondary care records when utilising this risk score with patients. Both the prior number of attacks and the time since the last attack will be included as predictors and will be considered time-dependent and accurate at the weekly level.

## Analysis plan

The derivation dataset (ALHS) will be divided into three partitions: 60% for training, 20% for model comparison (validation), and 20% to assess performance (testing). In our training subset, the first partition, we will train machine learning models (classifiers) with varying hyperparameters, predicting asthma attack occurrence in the following 1, 4, 26, and 52 weeks. We will run 100 iterations for statistical confidence, each time randomly permuting samples prior to determining the three subsets. The *no free lunch theorem* in machine learning suggests that there is no classifier (or more generically a machine learning tool) which will consistently outperform competing approaches across all settings.[67] Therefore, given that we do not know a priori which classifier will work best in this application, we will apply naïve Bayes classifiers for benchmarking and then employ more advanced state-of-the-art principled supervised learning algorithmic tools such as support vector machines, random forests, and ensembles (classifier combinations) to investigate which algorithm leads to more accurate results.

A selection of *training enrichment* methods will be trialled in order to assess how to best overcome poor performance as a result of low outcome prevalence. Typically, modelling rare events results in reduced sensitivity (the proportion of the individuals who had attacks that were detected), so the individuals predicted to be at low risk will have a high rate of asthma attacks. As such, the start of this process (the first 20 iterations of training each model) will be repeated five times using:

1. the original analysis dataset,
2. original data with additional duplicates of the positive outcome records (a method known as *over-sampling*),[68]
3. original data, with a selection of the negative outcome records removed (*under-sampling*),[68]
4. original data with additional slightly modified duplicates of the positive outcome records, with a selection of the negative outcome records removed (*synthetic minority over-sampling; SMOTE*),[68 69]
5. original data, using the outcome classification threshold to maximise the primary metric—the Matthew's correlation coefficient (MCC)[70]—identified using golden-section search optimisation.[71]

By assessing the average performance by classification method class, in each set of iterations, we will determine which enrichment method is the most appropriate overall for the data and to be continued accordingly.

In the validation partition, with all 100 iterations for the selected enrichment methods, we will identify the highest performing model as that with the highest mean MCC across iterations; in the event of a tie, the model with the highest iteration-minimum MCC will be selected.

Model testing will be conducted on the selected model (figure 1) in the derivation testing partitions. Model calibration will be assessed by comparing observed rate of

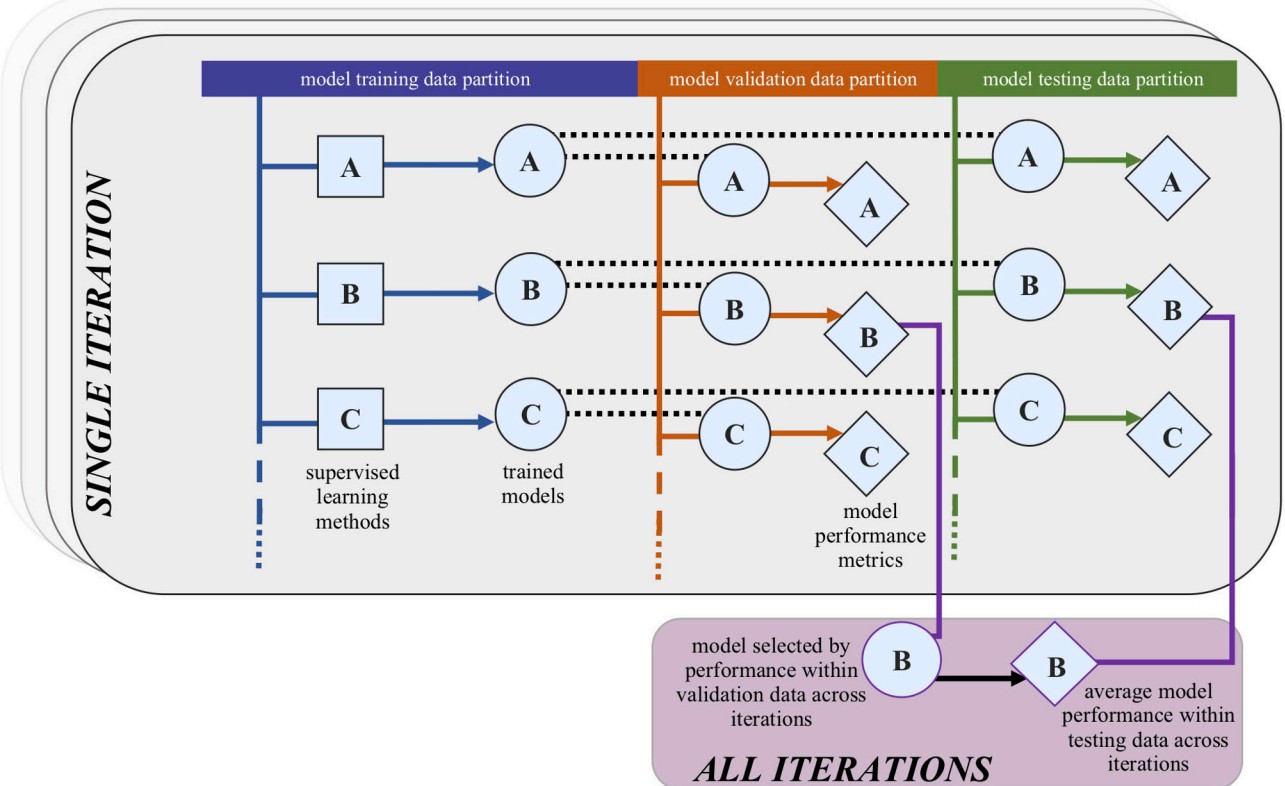

**Figure 1** Process of selecting the highest performing model from the validation data and the average performance of this model across iterations in the testing dataset. In the foreground, we have the first iteration. We will use 100 iterations for statistical confidence, randomly permuting the data into training, validation, and testing subsets in each iteration.

incidence by predicted risk for the full population and by exhaustive population subgroups, including asthma severity, prior number of asthma attacks, age, and smoking status (particularly useful to assess possible contamination by asthma-COPD overlap syndrome (ACOS)). We will also check the calibration between the predicted risk and the attack incidence, stratified by the source of the asthma attack record (in primary care, A&E presentation, or inpatient admission). Performance in the testing datasets will be assessed using the MCC, and the additional metrics of sensitivity, specificity, positive and negative predictive values, and the $F_1$ measure,[72] along with information criteria such as the Bayesian Information Criterion are calculated to obtain a trade-off between model complexity and accuracy. Confusion matrices (also known as contingency tables) will be made available as online supplementary materials.

The derivation dataset will be re-used in its entirety to retrain the model based on the final classifier and hyperparameter selection. Model testing will then be conducted in the external dataset, which consists of data unseen in the model derivation, using this trained model. Distributions of predictors between the derivation and external datasets will be assessed (indirectly) to contextualise the generalisability findings. The aforementioned metrics will be reported.

Finally, we will re-train the model using the hyperparameter specifications from the best performing model,

with a modified version of the derivation dataset which incorporates data extracted from secondary care records (such as A&E presentations for asthma attack not captured in primary care records) in the determination of the risk factors. This allows us to evaluate the added value of secondary care data linkage in the prediction of impending asthma attacks and will be determined by the same metrics used for the primary model evaluation.

All analyses will be conducted in R (though the RStudio interface), and details on the functions, the hyperparameter within each classifier, and the ranges assessed herein are provided in Appendix B.

### Ethics and dissemination

All authors with data access have completed the Safe Users of Research data Environment training, provided by the Administrative Data Research Network. All analysis will be conducted in concordance with the National Services Scotland Electronic Data Research and Innovation Service (eDRIS) user agreement. This study protocol will be registered with the European Union electronic Register of Post-Authorisation Studies (EU PAS Register) as a non-interventional post-authorisation study (PAS) before any data analysis is initiated.

The subsequent research paper will be submitted for publication in a peer-reviewed journal and will be written in accordance with TRIPOD: *transparent reporting of a multivariable prediction model for individual prognosis*

or diagnosis[73] and RECORD: *reporting of studies conducted using observational routinely-collected health data*[74] guidelines. Code scripts used for all components of the data cleaning, compiling, and analysis will be made available in the open source GitHub website at https://github.com/hollytibble.

A lay summary of this protocol paper, and the subsequent research results paper, will be made available online (via an open source platform) in order to heighten the impact and accessibility of this work. A lay summary on asthma will be provided as online supplementary materials.

## CONCLUSIONS

This project will further advance asthma attack risk prediction modelling and will inform on the future direction of routine data linkage in Scotland, which is likely to have additional benefits for other health systems in the UK and internationally.

**Acknowledgements**  The authors would like to thank Eleftheria Vasileiou, Amy Tilbrook, Mome Mukherjee, and Jill Tibble for their contributions to the analysis plan, data management, and proof-reading of this manuscript, and the Asthma UK Centre for Applied Research (AUKCAR) Patient and Public Involvement group for their contribution to the analysis plan.

**Contributors**  HT and AT conceived and planned the analysis. HT and RH specified the medication adherence measures. HT, EH, CS, MM, and AS constructed the covariate (and associated Read Coding) lists for the model. HT wrote the first draft, with contributions from all authors. All authors (HT, AT, EH, RH, MM, CRS, and AS) approved the final version and jointly take responsibility for the decision to submit this manuscript to be considered for publication.

**Funding**  HT is supported by College of Medicine and Veterinary Medicine PhD (eHERC/Farr Institute) Studentships from The University of Edinburgh. EH is supported by a Medical Research Council PhD Studentship (eHERC/Farr). MAM's Newton International Fellowship is awarded by the Academy of Medical Sciences and Newton Fund. This work is carried out with the support of the Asthma UK Centre for Applied Research [AUK-AC-2012-01] and Health Data Research UK, an initiative funded by UK Research and Innovation Councils, National Institute for Health Research (England) and the UK devolved administrations, and leading medical research charities. The ALHS dataset was created with funding from the National Environment Research Council [NE/P011012/1]. The SIVE II dataset was created with funding from the National Institute for Health Research (NIHR) Health Technology Assessment programme [13/34/14]—the views and opinions expressed therein are those of the authors and do not necessarily reflect those of the Health Technology Assessment programme, NIHR, NHS, or the Department of Health.

**Competing interests**  None declared.

**Patient consent for publication**  Not required.

**Ethics approval**  Permissions for the ALHS project were obtained from the South East Scotland Research Ethics Committee 02 [16 /SS/0130] and the Public Benefit and Privacy Panel for Health and Social Care (1516 – 0489) . Permissions for the SIVE II project were obtained from the Privacy Advisory Committee (National Services NHS Scotland) [68/14] and the National Research Ethics Committee West Midlands - Edgbaston [15/WM/0035].

**Provenance and peer review**  Not commissioned; externally peer reviewed.

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
