## [Reviewer comments · BMJ Open]

ARTICLE DETAILS

TITLE (PROVISIONAL)	Predicting asthma attacks in primary care: protocol for developing a machine learning-based prediction model
AUTHORS	Tibble, Holly; Tsanas, Athanasios; Horne, Elsie; Horne, Robert; Mizani, Mehrdad; Simpson, Colin; Sheikh, Aziz

VERSION 1 - REVIEW

REVIEWER	Abdel Douiri King's College London. United Kingdom
REVIEW RETURNED	14-Feb-2019

GENERAL COMMENTS	The topic and proposed research methodology are interesting for those in asthma research, however, it would be helpful if the authors could address the points as follows: First, the research objectives are not clearly written. It seems like the researchers were trying to develop a risk prediction model in order to identify patients at high risk of asthma attack. But in the manuscript, high specificity of the model to avoid unnecessary prescription for prophylactic purposes was also mentioned. There is a trade-off between sensitivity and specificity for single prediction model. If high sensitivity is expected to avoid fatal cases as indicated by this protocol, it would be confusing to keep mentioning high specificity in the abstract as well as the introduction section. Second, asthma attack comes up with worsening of asthma symptoms within a short time especially for severe asthma attack. Such sudden onset of asthma attacks results in some fatal cases without timely access to medical help. This raises a question of the feasibility of using routine collected data to develop prediction model for urgent conditions. Even if in the manuscript the authors referenced a study by Finkelstein and Jeong, in which tele-monitoring data was adopted for asthma attack prediction modelling, it is still not convincing that the improvements in sensitivity was due to the novelty of methodology rather inherently rooted in the nature of the data being monitoring data. Admittedly modelling approaches excise an essential role in disease prediction research, health information being employed remains to be fundamental in this type of research. Please provide enough rationale that using primary care data as main information resource could predict a health outcome characterised with rapid progression. Also, it would be helpful to specify whether data linkage to A&E and secondary care records serves for case
--

	identification or adding up information to primary care health data. Alternatively, it would be useful to refine the research question within a feasible range i.e. stratifying the severity of asthma attack if possible. Third, the authors did notice the issue of class imbalance and its effects on prediction model performance. However, in the manuscript, there is no information on the incidence/ baseline rates regarding asthma attacks among asthma patients to support the argument of class imbalance and subsequent strategies for overcoming it in their analysis plan. Fourth, the authors did not address their choice of modelling techniques adequately. Given the main purpose of this research is not focusing on testing modelling approaches neither exploring statistical methodology, the authors should state why they explicitly chose random forests, naïve Bayes, SVN and their combination rather than other statistical modelling techniques or machine learning approaches, since significant differences between results generated by proposed modelling methods are generally not expected. The author could simply choose one machine learning method plus a regression-based statistic with the later quantifying the average contribution of individual predictors as complementary information. Fifth, it would be clear if the authors provide a definition of 'raw data' as mentioned in the analysis plan.
--	--

REVIEWER	Steve Turner
	University of Aberdeen, UK
REVIEW RETURNED	03-Mar-2019

GENERAL COMMENTS	I will start by congratulating the authors for conceiving this study and presumably obtaining funding to execute the study. Here are a few points which I believe will improve the reader's understanding of what is proposed:  1. Introduction (page 5 of 20, lines 30-40). This paragraph confuses severity, control and exacerbation, a clear message could be that of all the factors tested, exacerbation is the most reliable predictor of exacerbation. 2. Introduction (page 5 of 20, lines 52-59). The authors could more fully explain to the non-specialist reader (such as me) how the proposed statistical approach improves sensitivity, perhaps giving an example. Also the authors should state how the results of the analysis are of clinical benefit to the patient (does the analysis yield a personalised risk factor that could be part of dynamic risk assessment?) 3. Methods. What is the benefit of linking LHS to SIVE II? Is this just to add more individuals to the analysis? 4. PPI. This does not seem to be relevant to the proposed study where the focus is predicting exacerbations. The PPI focus is reducing the burden of steroid prescribing. 5. Inclusion criteria. Are there any age cut offs? Asthma diagnosis under the age of five can be under certain and over the age of 40 can be confused with COPD. Some a priori exclusion criteria should be stated, for example are there some co-morbidities? 6. Blood eosinophil count. I do not think it is valid to assume that individuals with missing eosinophil count do not have eosinophilia.
--

	An asthmatic population will be enriched with individuals who have eosinophilia. To me missing values should be assigned missing, the study population should have no problem with power. 7. Obesity. In children it is not valid to define obesity based on a BMI cut off. 8. Analysis plan. How will the authors consider the relapse/remission pattern of asthma? For example asthma control is likely to vary on a year-by-year basis. At present the plan is for characteristics at baseline to be used as confounders, but given the natural ebb and flow of asthma over time if the authors do not consider how factors change over time (e.g. control, ICS dose, the use of any add on treatment, smoking cessation, adherence, etc) then their model will lose precision.
--	---

VERSION 1 – AUTHOR RESPONSE

Response to Reviewer 1:

Thank you, Dr. Douiri, for your detailed and thoughtful comments, we hope our amendments address your concerns. We felt that these changes were particularly useful to consolidate our narrative and improve comprehension.

1. The research objectives are not clearly written. It seems like the researchers were trying to develop a risk prediction model in order to identify patients at high risk of asthma attack. But in the manuscript, high specificity of the model to avoid unnecessary prescription for prophylactic purposes was also mentioned. There is a trade-off between sensitivity and specificity for single prediction model. If high sensitivity is expected to avoid fatal cases as indicated by this protocol, it would be confusing to keep mentioning high specificity in the abstract as well as the introduction section.

We have added the following sentence at the end of the first paragraph in the section Research Aim, which we believe clarifies our position:

“The model will build on previous research 1–9 to improve the sensitivity of our event prediction, without unduly compromising the specificity. This is crucial in order to reduce steroid prescribing and diminish the long-term effects of high steroid use over a life time, which have adverse effects 10–18, and reduce patient anxiety when risk of an asthma attack is low.”

2. Asthma attack comes up with worsening of asthma symptoms within a short time especially for severe asthma attack. Such sudden onset of asthma attacks results in some fatal cases without timely access to medical help. This raises a question of the feasibility of using routine collected data to develop prediction model for urgent conditions. Even if in the manuscript the authors referenced a study by Finkelstein and Jeong, in which tele-monitoring data was adopted for asthma attack prediction modelling, it is still not convincing that the improvements in sensitivity was due to the novelty of methodology rather inherently rooted in the nature of the data being monitoring data.

Admittedly modelling approaches excise an essential role in disease prediction research, health information being employed remains to be fundamental in this type of research. Please provide enough rationale that using primary care data as main information resource could predict a health outcome characterised with rapid progression. Also, it would be helpful to specify whether data linkage to A&E and secondary care records serves for case identification or adding up information to primary care health data. Alternatively, it would be useful to refine the research question within a feasible range i.e. stratifying the severity of asthma attack if possible.

We have added the following new text under the section Research Aim to further justify the appropriateness of using routinely collected data:

“Primary care consultations provide the opportunity for patients and clinicians to assess changes to asthma attack risk, which can be used to promote patients to seek emergency care if there is a significant deterioration in their symptoms, and to promote risk-reducing lifestyle choices.”

Regarding the Finkelstein and Jeong study, we cannot be certain that training data enrichment will consistently improve sensitivity, however for each method the authors used, the addition of training data enrichment increased the sensitivity. We will be comparing each of our data enrichment methods to the original data as a control dataset (used as benchmark), in order to assess whether we also find this consistent improvement. If our results differ, that will certainly be an interesting finding.

Regarding data linkage, we have removed the following sentence from the abstract, as we agree that this wording was not sufficiently clear, but did not feel it could be greatly improved within the constraints of the abstract word allowance:

“We will investigate the potential added value across various metrics (including sensitivity and specificity) by extending the statistical learning model, incorporating information extracted from linked secondary care records in addition to the primary care EHR data.”

We have amended the beginning of the final paragraph in the Analysis Plan section, as follows:

“Finally, we will re-train the model using the hyperparameter specifications from the best performing model, with a modified version of the derivation dataset which incorporates data extracted from secondary care records (such as A&E presentations for asthma attack not captured in primary care records) in the determination of the risk factors. This allows us to evaluate the added value of secondary care data linkage in the prediction of impending asthma attacks, and will be determined by the same metrics used for the primary model evaluation.”

We hope this clarifies that hitherto, secondary care records had only been used to ascertain clinical outcomes, and not as predictors used in the determination of previous asthma attacks. In this

additional analysis we will assess how using a combination of primary and secondary care records to ascertain the risk factors might improve risk prediction performance.

Finally, we have added the following sentence to the 5th of the Analysis Plan section:

“We will also check the calibration between the predicted risk and the attack incidence, stratified by the source of the asthma attack record (in primary care, A&E presentation, or inpatient admission).”

3. The authors did notice the issue of class imbalance and its effects on prediction model performance. However, in the manuscript, there is no information on the incidence/ baseline rates regarding asthma attacks among asthma patients to support the argument of class imbalance and subsequent strategies for overcoming it in their analysis plan.

We have added the following sentence at the end of the first paragraph in the Introduction section:

“Asthma attack incidence is reported to be between 0.01 and 0.78 events per person-year, depending on the definition of attacks, and the population (e.g. primary care, secondary care) 5,19,20.”

This sets the scene at the very beginning that this is a problem that will have class imbalances. This is addressed later in the Introduction in the paragraph beginning with ‘Despite the identification...’, where we have amended a sentence to now read:

“Logistic regression, the most commonly used statistical method for event prediction, is known to predict outcomes poorly when there is class imbalance (event and no event) 21, and we expect the problem investigated in this study assessing asthma attacks will be highly imbalanced.”

4. The authors did not address their choice of modelling techniques adequately. Given the main purpose of this research is not focusing on testing modelling approaches neither exploring statistical methodology, the authors should state why they explicitly chose random forests, naïve Bayes, SVN and their combination rather than other statistical modelling techniques or machine learning approaches, since significant differences between results generated by proposed modelling methods are generally not expected. The author could simply choose one machine learning method plus a regression-based statistic with the later quantifying the average contribution of individual predictors as complementary information.

We have modified the final sentence of the first paragraph of the analysis plan to read as follows:

“The no free lunch theorem in machine learning suggests there is no classifier (or more generically a machine learning tool) which will consistently outperform competing approaches across all settings 22. Therefore, given that we do not know a priori which classifier will work best in this application, we will apply naïve Bayes classifiers for benchmarking, and then employ more advanced state of the art principled supervised learning algorithmic tools such as support vector machines, random forests, and ensembles (classifier combinations) to investigate which algorithm leads to more accurate results.”

5. It would be clear if the authors provide a definition of ‘raw data’ as mentioned in the analysis plan.

The term ‘raw data’ appears in the engineering and data analysis literature, but we appreciate the reviewer’s comment that this might be confusing to readers in a clinical journal. Therefore, we have changed ‘raw’ to ‘original’ in the second paragraph of the analysis plan section, which we hope clarifies that it is the cleaned analysis dataset.

Response to Reviewer 2:

Thank you, Dr. Turner, for your detailed enquiries and suggestions about the protocol paper, as well as your encouraging words. We hope these amendments address satisfactorily the comments raised, and help to better elucidate our rationale.

1. Introduction (page 5 of 20, lines 30-40). This paragraph confuses severity, control and exacerbation, a clear message could be that of all the factors tested, exacerbation is the most reliable predictor of exacerbation.

We have revised the wording of paragraph 4 in the introduction, as follows, which we hope simplifies our argument, and avoids the conflation of control with severity:

“While it might seem intuitive that those with the most severe daily symptoms exhibit greater risk of severe morbidity and mortality, research suggests that these symptoms may be a suboptimal clinical marker of asthma attack risk 23. Indeed, some people with asthma are more prone to attacks than others, with past attack history being the strongest risk factor for future attacks 24–27.”

2. Introduction (page 5 of 20, lines 52-59). The authors could more fully explain to the non-specialist reader (such as me) how the proposed statistical approach improves sensitivity, perhaps giving an example. Also the authors should state how the results of the analysis are of clinical benefit to the patient (does the analysis yield a personalised risk factor that could be part of dynamic risk assessment?)

Regarding the benefit to sensitivity, the strength in our methods lies in the use of training data enrichment methods. We have added an additional sentence, as follows, to the 5th paragraph of the introduction, we hope helps to clarify the relationship between rare events and poor sensitivity, in order to lead into the introduction to these enrichment methods:

“For example, a model could predict that a very rare event would never occur, and it would be correct in the vast majority of cases.”

Regarding the clinical benefit of this study, we have revised the wording in the first sentence of the Research Aim to better phrase how the risks core could be used in clinical care as a preventative care decision support tool, as follows:

“We aim to create a personalised risk assessment tool to assist primary care clinicians in anticipating asthma attacks in the following 1, 4, 12, 26, and 52 weeks, employing machine learning methodologies such as naïve Bayes classifiers, random forests, and support vector machines, as well as ensemble algorithms.”

3. What is the benefit of linking LHS to SIVE II? Is this just to add more individuals to the analysis?

We will not be conducting a data linkage between the ALHS and SIVE II datasets. The SIVE II dataset will be used as an external validation set in order to demonstrate that any good results (high Matthew's Correlation Coefficient) is not limited to the specific dataset that we developed the model in, and that it is appropriate to use generally in the Scottish primary care setting.

We have revised the first sentence in the second paragraph of the Data sources and permissions section, as follows, which we believe clarifies this:

“In order to verify that the prediction model performance is not limited to the development dataset, and that it generalizes well in new, unseen data presented to the classifier in the training process, we will evaluate its performance using an external cohort study dataset, the second Seasonal Influenza Vaccination Effectiveness (SIVE II) cohort study...”

4. PPI. This does not seem to be relevant to the proposed study where the focus is predicting exacerbations. The PPI focus is reducing the burden of steroid prescribing.

We have amended this section to better reflect avoiding compromising in terms of specificity, rather than increasing the (already high) specificity, as follows:

“The particular importance of avoiding a substantial decrease in specificity in order to gain higher sensitivity was a result of discussions within this group about the burden of side-effects from preventative treatment.”

5. Are there any age cut offs? Asthma diagnosis under the age of five can be under certain and over the age of 40 can be confused with COPD. Some a priori exclusion criteria should be stated, for example are there some co-morbidities?

We have amended the inclusion criteria to adults without a COPD diagnosis only, as noted in the following revised first sentence of the Inclusion criteria section:

“We will identify our study population as all adults (aged 18 and over) with asthma identified by clinical diagnoses (Read codes), without a COPD diagnosis, and with relevant prescribing records in primary care.”

We decided to refine the population to specifically adults, rather than those over the age of 5, as we felt that when assessing risk in children and young people

it would be pertinent to include additional information about their caregivers 24, which was not available to us. We decided not to place an upper limit on the age of participants, but we will conduct calibration assessments by age. We have added the following new text in the 5th paragraph of the Analysis Plan section:

“Model calibration will be assessed by comparing observed rate of incidence by predicted risk, for the full population and by exhaustive population subgroups, including asthma severity, prior number of asthma attacks, age and smoking status (particularly useful to assess possible contamination by asthma-COPD overlap syndrome (ACOS)).”

There are no further exclusion criteria, and co-morbidities will be included as covariates.

6. I do not think it is valid to assume that individuals with missing eosinophil count do not have eosinophilia. An asthmatic population will be enriched with individuals who have eosinophilia. To me missing values should be assigned missing, the study population should have no problem with power.

We have amended the coding levels as suggested, within the Patient characteristics, confounders, and missing data handling section:

“Those with non-recorded blood eosinophil count will be coded as missing.”

7. In children it is not valid to define obesity based on a BMI cut off.

We have amended the inclusion criteria to adults only, as noted in the revised first sentence of the Inclusion criteria section:

“We will identify our study population as all adults (aged 18 and over) with asthma identified by clinical diagnoses (Read codes), without a chronic obstructive pulmonary disease (COPD) diagnosis, and with relevant prescribing records in primary care.”

Therefore, we do not need to discuss BMI cut-off for children.

8. How will the authors consider the relapse/remission pattern of asthma? For example asthma control is likely to vary on a year-by-year basis. At present the plan is for characteristics at baseline to be used as confounders, but given the natural ebb and flow of asthma over time if the authors do not consider how factors change over time (e.g. control, ICS dose, the use of any add on treatment, smoking cessation, adherence, etc) then their model will lose precision.

We have amended the wording of the first sentence in the Patient characteristics, confounders, and missing data handling section to clarify that covariates such as asthma severity, smoking status, and healthcare usage are not static – but instead time-dependent (or time varying) as stated subsequently within the subsections of each of these variables:

“Patient characteristics at baseline will be reported, and included as time-varying confounders in analyses.”

References within modified or removed manuscript statements

1. Loymans, R. J. B. et al. Identifying patients at risk for severe exacerbations of asthma: development and external validation of a multivariable prediction model. *Thorax* 71, 838–846 (2016).
2. Schatz, M., Cook, E. F., Joshua, A. & Petitti, D. Risk Factors for Asthma Hospitalizations in a Managed Care Organization: Development of a Clinical Prediction Rule. *Am. J. Manag. Care* 9, 538–547 (2003).

3. Lieu, T. A., Quesenberry, C. P., Sorel, M. E., Mendoza, G. R. & Leong, A. B. Computer-based models to identify high-risk children with asthma. *Am J Respir Crit Care Med* (1998).
4. Smith, J. R. et al. The at-risk registers in severe asthma (ARRISA) study: A cluster-randomised controlled trial examining effectiveness and costs in primary care. *Thorax* 67, 1052–1060 (2012).
5. Loymans, R. J. B. et al. Exacerbations in Adults with Asthma: A Systematic Review and External Validation of Prediction Models. *J. Allergy Clin. Immunol. Pract.* (2018).
6. Van Vliet, D. et al. Prediction of asthma exacerbations in children by innovative exhaled inflammatory markers: Results of a longitudinal study. *PLoS One* 10, 1–15 (2015).
7. Hallit, S. et al. Development of an asthma risk factors scale (ARFS) for risk assessment asthma screening in children. *Pediatr. Neonatol.* (2018).
8. Forno, E. et al. Risk factors and predictive clinical scores for asthma exacerbations in childhood. *Chest* 138, 1156–1165 (2010).
9. Finkelstein, J. & Jeong, I. cheol. Machine learning approaches to personalize early prediction of asthma exacerbations. *Ann. N. Y. Acad. Sci.* 1387, 153–165 (2017).
10. Kim, S. Y. et al. Incidence and Risk Factors of Steroid-induced Diabetes in Patients with Respiratory Disease. *J Korean Med Sci* 26, 264–267 (2011).
11. Suissa, S., Kezouh, A. & Ernst, P. Inhaled Corticosteroids and the Risks of Diabetes Onset and Progression. *AJM* 123, 1001–1006 (2010).
12. Blackburn, D., Hux, J. & Mamdani, M. Quantification of the risk of corticosteroid-induced diabetes mellitus among the elderly. *J. Gen. Intern. Med.* 17, 717–720 (2002).
13. Adinoff, A. D. & Hollister, J. R. Steroid-Induced Fractures and Bone Loss in Patients with Asthma. *N. Engl. J. Med.* 309, 265–268 (1983).
14. Van Staa, T. P., Leufkens, H. G., Abenhaim, L., Zhang, B. & Cooper, C. Use of oral corticosteroids and risk of fractures. *J Bone Min. Res* (2000).
15. Dawson, K. L. & Carter, E. R. A steroid-induced acute psychosis in a child with asthma. *Pediatr. Pulmonol.* 26, 362–364 (1998).
16. Kayani, S. & Shannon, D. C. Adverse behavioral effects of treatment for acute exacerbation of asthma in children: A comparison of two doses of oral steroids. *Chest* 122, 624–628 (2002).
17. Brown, E. S., Khan, D. A. & Nejtek, V. A. The psychiatric side effects of corticosteroids. *Ann. Allergy, Asthma Immunol.* 83, 495–504 (1999).
18. Bloechliger, M. et al. Adverse events profile of oral corticosteroids among asthma patients in the UK: cohort study with a nested case- control analysis. *Respir. Res.* 19, (2018).
19. Royal College of Physicians. Why asthma still kills: The National Review of Asthma Deaths (NRAD). (2014).
20. Mukherjee, M., Nwaru, B. I., Soyiri, I., Grant, I. & Sheikh, A. High health gain patients with asthma: a cross-sectional study analysing national Scottish data sets. *Prim. Care Respir. Med.* 28, 27 (2018).

21. King, G., Zeng, L. & King, G. Logistic Regression in Rare Events Data. *Polit. Anal.* 9, 137–163 (2001).
22. Wolpert, D. H. & Macready, W. G. No free lunch theorems for optimization. *IEEE Trans. Evol. Comput.* 1, 67–82 (1997).
23. Green, R. H., Brightling, C. E. & McKenna, S. Asthma exacerbations and eosinophil counts. A randomised controlled trial. *Lancet* 360, 1715–21 (2002).
24. Buelo, A. et al. At-risk children with asthma (ARC): a systematic review. *Thorax* 01136, 1–12 (2018).
25. Turner, M. O. et al. Risk factors for near-fatal asthma A case-control study in hospitalized patients with asthma. *Am. J. Respir. Crit. Care Med.* 157, 1804–1809 (1998).
26. ten Brinke, A. et al. Risk factors of frequent exacerbations in difficult-to-treat asthma. *Eur. Respir. J.* (2005).
27. Turner, S. W., Murray, C., Thomas, M., Burden, A. & Price, D. B. Applying UK real-world primary care data to predict asthma attacks in 3776 well-characterised children: a retrospective cohort study. *npj Prim. Care Respir. Med.* 28, 28 (2018).

VERSION 2 – REVIEW

REVIEWER	Steve Turner University of Aberdeen, UK
REVIEW RETURNED	04-Apr-2019

GENERAL COMMENTS	Nice revision. Very good luck with the study.
---